# Whole-Genome Analysis Reveals the Dynamic Evolution of Holocentric Chromosomes in Satyrine Butterflies

**DOI:** 10.3390/genes14020437

**Published:** 2023-02-08

**Authors:** Elena A. Pazhenkova, Vladimir A. Lukhtanov

**Affiliations:** 1Department of Biology, Biotechnical Faculty, University of Ljubljana, Večna pot 111, 1000 Ljubljana, Slovenia; 2Department of Karyosystematics, Zoological Institute of Russian Academy of Sciences, Universitetskaya nab. 1, St. Petersburg 199034, Russia

**Keywords:** chromosomal speciation, chromosome-scale genome assembly, *Erebia*, genome, holocentromere, karyotype, Lepidoptera, *Maniola*, Nymphalidae, Z chromosome

## Abstract

Butterfly chromosomes are holocentric, i.e., lacking a localized centromere. Potentially, this can lead to rapid karyotypic evolution through chromosome fissions and fusions, since fragmented chromosomes retain kinetic activity, while fused chromosomes are not dicentric. However, the actual mechanisms of butterfly genome evolution are poorly understood. Here, we analyzed chromosome-scale genome assemblies to identify structural rearrangements between karyotypes of satyrine butterfly species. For the species pair *Erebia ligea*–*Maniola jurtina*, sharing the ancestral diploid karyotype 2n = 56 + ZW, we demonstrate a high level of chromosomal macrosynteny and nine inversions separating these species. We show that the formation of a karyotype with a low number of chromosomes (2n = 36 + ZW) in *Erebia aethiops* was based on ten fusions, including one autosome–sex chromosome fusion, resulting in a neo-Z chromosome. We also detected inversions on the Z sex chromosome that were differentially fixed between the species. We conclude that chromosomal evolution is dynamic in the satyrines, even in the lineage that preserves the ancestral chromosome number. We hypothesize that the exceptional role of Z chromosomes in speciation may be further enhanced by inversions and sex chromosome–autosome fusions. We argue that not only fusions/fissions but also inversions are drivers of the holocentromere-mediated mode of chromosomal speciation.

## 1. Introduction

In contrast to monocentric chromosomes, in which centromeres are restricted to single regions on each chromosome, holocentric chromosomes are characterized by long centromeric areas (i.e., holocentromeres) [1,2,3,4]. Holocentromeres are distributed along the poleward surface of metaphase chromosomes, such that during cell divisions, microtubules bind to the chromosomes along almost their entire length [5]. The most recent study of holocentric chromosomes in beak-shaped species (*Rhynchospora*, family Cyperaceae) showed that these chromosomes are in fact polycentric [4], which significantly modifies previous models of holocentromere organization and indicates the need for a more detailed study of their structures in different organisms.

One of the most remarkable features of organisms with holocentric chromosomes is their ability to tolerate the fragmentation and fusion of chromosomes. In monocentric organisms, all chromosomal rearrangements must comply with the persistence of a functional centromere [6]; otherwise, they are not viable. For example, in monocentric organisms, fragmentation causes the formation of centric and acentric fragments, and fusion can cause the formation of dicentric chromosomes. In cell divisions, acentric fragments do not segregate and are consequently lost [7], whereas dicentric chromosomes are mostly broken [8]. In contrast, in holocentric organisms, fragmented chromosomes retain kinetic activity, and fused chromosomes are not dicentric and do not break during cell division [3,7].

Another remarkable feature found in many groups of holocentric organisms [9,10,11,12,13,14] is inverted meiosis. In standard conventional meiosis, the first meiotic division is reductional, resulting in the segregation of homologous chromosomes, whereas the second meiotic division is equational, resulting in the separation of sister chromatids. In inverted meiosis, the opposite order of these main meiotic events occurs, and homologous chromosomes separate after sister chromatids [15,16,17,18]. For butterflies of the genus *Leptidea* (Lepidoptera, family Pieridae), it has been shown that inverted meiosis facilitates proper chromosome segregation and rescues fertility and viability in hybrids heterozygous for chromosomal fusions and fissions [19,20]. Both the tolerance to chromosomal fusions and fragmentations and chromosomal heterozygosity are expected to favor the fixation of novel chromosomal rearrangements [3,7]. Therefore, holocentromeres have been hypothesized to drive rapid karyotype evolution [7,21].

This hypothesis, despite its popularity, has not yet received either reliable confirmation or refutation. In particular, comparative phylogenetic analysis has shown that chromosome number seems to evolve at equal rates in holocentric and monocentric clades [22]. There are numerous examples that demonstrate both the unusually high and unexpectedly conservative evolution of holocentric karyotypes. Thus, in the holocentric insect order Lepidoptera, which includes more than 150,000 species, in the blue butterfly genus *Polyommatus* (Lepidoptera, family Lycaenidae), in less than 5 million years of its evolution [23], a fan of chromosomally differentiated species arose. In these species, the diploid numbers (2*n*) of chromosomes vary from 2*n* = 20 to 2*n* = 452 [24,25,26]. A comparable level of interspecific variability in chromosome numbers is known for the satyrine genus *Erebia* (Lepidoptera, Nymphalidae, Satyrinae) (the diploid numbers vary from 2*n* = 14 to 2*n* = 104) and the tribe Ithomiini (Lepidoptera, family Nymphalidae) (the diploid numbers vary from 2*n* = 10 to 2*n* = 240) [27,28,29]. It would seem that these examples support the hypothesis that holocentricity accelerates karyotypic evolution. However, the vast majority of karyotyped Lepidoptera species demonstrate conservatism in chromosome numbers. Most species of Lepidoptera have a haploid number (*n*) of chromosomes, *n* = 31 [28]. This fundamental genome feature is ancestral to the order Lepidoptera [30,31] and has been preserved in most families for more than 100 million years of their evolution [32,33].

However, the stability of chromosome numbers is weak evidence of the stability of karyotypes, since chromosomal rearrangements, both intrachromosomal (e.g., inversions) and interchromosomal (e.g., translocations), can lead to the total reorganization of the gene order without changing the number of chromosomes. The high interspecies variability in the number of chromosomes is a good indication of rapid chromosomal evolution. Despite this, a simple count of the number of chromosomes is an unreliable tool for the analysis of the patterns and mechanisms of rapid karyotypic evolution. The chromosome number itself usually does not provide information on how individual chromosomes and the whole karyotypes originated or how they are related to chromosome complements of other species [6].

In the era of classical cytogenetics in the 20th century, for methodological reasons, the study of Lepidoptera karyotypes was mainly limited to counting numbers of chromosomes, mostly in meiotic metaphase plates [27,28,29,30,31]. The analytical tools of the 21st century (BAC-FISH mapping and whole-genome analyses) reveal chromosomal rearrangements, but so far, few species of butterflies and moths have been studied to the extent that allows the comparison of karyotypes [34,35,36,37,38,39,40]. The results of these studies are contradictory. The presence of multiple translocations between species with similar numbers of chromosomes has been directly demonstrated for the butterfly family Pieridae [38]. Multiple chromosomal rearrangements were detected between species and populations of the genus *Leptidea* (Pieridae) with different chromosome numbers [41]. Despite this, genomic data for other butterflies and moths (Lepidoptera) show a high level of chromosomal synteny when comparing chromosomes of different non-closely related species [33,36,42,43,44,45,46]. For example, extensive conserved synteny of genes was found between the karyotypes of *Manduca sexta* and *Bombyx mori*, belonging to two different moth families (Sphingidae and Bombycidae). Only a few rearrangements, including three inversions, three translocations and two fission/fusion events, were estimated to have occurred after the divergence of these families [47]. At the same time, *M. sexta* shows a high level of intraspecific differentiation, with the Arizona population being differentiated from other populations by two large inversions [48].

In the species-rich group of satyrine butterflies (Lepidoptera, subfamily Satyrinae of the family Nymphalidae), the modal and probably ancestral karyotype is 2*n* = 58, as is found in many genera [28]. In the genus *Erebia*, which belongs to the subfamily Satyrinae, there is a clear tendency toward a decrease in the diploid number of chromosomes up to 2*n* = 14 [49], against the background of the preservation of the ancestral 2*n* = 58 in many species of the genus [28]. The chromosomal rearrangements that may be behind these superficially similar or highly altered satyrine karyotypes remain unknown. In our work, we conducted a comparative analysis of genome-wide assemblies for three pairs of satyrine species. The first pair is represented by the species *E. ligea* and *M. jurtina* and demonstrates the preservation of the ancestral haploid chromosome number *n* = 29. The second pair is represented by the species *E. aethiops* and *M. jurtina*, demonstrating different chromosome numbers in different genera. The third pair is represented by the species *E. ligea* and *E. aethiops*, demonstrating a significant decrease in chromosome number within the same genus.

## 2. Materials and Methods

### 2.1. Materials

Chromosome-level genome assemblies of *M. jurtina* (Linnaeus, 1758), *E. ligea* (Linnaeus, 1758) and *E. aethiops* (Esper, 1777) (Insecta, Lepidoptera, Nymphalidae, Satyrinae) generated by the Darwin Tree of Life Project [50,51,52,53] and freely available upon their deposition in the European Nucleotide Archive (ENA) (
s://www.darwintreeoflife.org/wp-content/uploads/2020/03/DToL-Open-Data-Release-Policy-1.pdf, accessed on 30 January 2023) were used for the analysis of chromosomal macrosynteny and collinearity and the detection of chromosomal rearrangements (Table 1 and Appendix A).

### 2.2. Detection of Macrosynteny and Chromosomal Rearrangements

We detected macrosynteny in the studied species through the pairwise whole-genome alignment of chromosome-level assemblies using minimap2 with the “asm20” preset, allowing higher divergence between genomes [54]. The advantages of this analysis are that it not only reveals syntenic regions but also visualizes chromosomal changes in terms and pictures of classical cytogenetics, i.e., provides graphic representations of chromosomes and chromosomal rearrangements. For each pair of species, this method made it possible to obtain multiple nucleotide alignments with varying lengths (in our case, up to 52 kb). Each such alignment was interpreted as a syntenic chromosomal block. We discarded alignments with a mapping quality of less than 60 and a length of less than 2 kb and visualized them as pairwise genomic dot plots with the pafr R package [55]. Pairwise comparisons were inspected visually. Karyotypes were drawn and colored using the RIdeogram R package [56]. At present, the circular plot is the standard for presenting comparative genomic data [57]. Therefore, we also visualized the pairwise synteny comparison as a circular layout (circos plot) showing the correspondence of each block between species pairs with the circlize R package [58].

## 3. Results

### 3.1. Analysis of Macrosynteny and Collinearity

The pairwise comparisons between the species were performed using the minimap2 algorithm [54] and revealed multiple syntenic blocks of nucleotides; the length of the detected blocks varied from 2 kb to 52 kb (Table 2). Within each compared pair of chromosomes, these syntenic blocks mainly formed continuous sequences (shown as diagonals in the genomic dot plots, Figure 1, Figure 2 and Figure 3). In most cases, the length of these diagonals and the order of blocks in them corresponded to the length and order of blocks in the compared chromosomes. This indicates not only the macrosynteny of the compared chromosomes (and, consequently, their homology) but also their collinearity.

Pairwise comparisons were also performed using a circus plot analysis [57]. This approach revealed the same patterns of relationships between species and chromosomes (Figure 4) as the minimap2 algorithm [54]. In the circus plot analysis, chromosomes of *E. ligea* and *M. jurtina* were found to be syntenic. Chromosomes 1–8, 18 and neo-Z of *E. aethiops* were found to be formed as a result of interchromosomal fusions.

### 3.2. Chromosomal Rearrangements: Inversions and Fusions

In the species pair *M. jurtina–E. ligea*, in eight autosomes and in the Z sex chromosome, the pairwise comparison revealed regions in which the sequence of syntenic blocks had the reverse orientation, thus indicating the presence of inversions (Figure 1). The inversions were located in subtelomeric (chromosomes 2, 6 22, 25 and Z of *E. ligea*) or interstitial (chromosomes 11, 17, 23 and 26 of *E. ligea*) positions. We did not find any interchromosomal rearrangements (translocations) between *M. jurtina* and *E. ligea*.

The comparison of *E. ligea* with *E. aethiops* shows that these taxa are separated by nine simple autosomal fusions (two smaller chromosomes of *E. ligea* correspond to a larger chromosome of *E. aethiops*) and the autosome 12–sex chromosome Z fusion resulting in a neo-Z chromosome in *E. aethiops* (Figure 3 and Figure 5). We also detected a single small subterminal inversion separating the Z chromosomes of this species (Figure 5). This is a different inversion than that found in the *M. jurtina–E. ligea* pair, since it has an almost terminal position, while in the *M. jurtina–E. ligea* pair, the inversion in the Z chromosome has a more interstitial localization.

The comparison between *M. jurtina* and *E. aethiops* (Figure 2) revealed nine autosomal fusions and one autosome–sex chromosome Z fusion. Taking into account the comparison between *E. ligea* and *E. aethiops* (Figure 3), we can conclude that all of these chromosome fusions arose in the lineage leading to the species *E. aethiops*, after its separation from *E. ligea*.

Since no inversions have been identified between *E. ligea* and *E. aethiops*, except for one terminal Z chromosome inversion (Figure 3), we can expect that all eight autosomal inversions and one subterminal Z chromosome inversion between *M. jurtina* and *E. ligea* (Figure 1) accumulated during evolution before the separation of *E. ligea* and *E. aethiops*. Therefore, one would expect that the same nine inversions (eight autosomal and one subterminal Z chromosome) would be found in the comparison between *M. jurtina* and *E. aethiops*. In fact, in the *M. jurtina–E. aethiops* comparison, seven inversions were revealed (Figure 2), which were identical to the inversions in the *M. jurtina* and *E. ligea* pair (in chromosomes 4*_jurtina_*, 7*_jurtina_*, 8*_jurtina_*, 21*_jurtina_*, 23*_jurtina_*, 25*_jurtina_* and Z*_jurtina_*). In the comparison between *M. jurtina* and *E. aethiops*, the expected inversions in chromosomes 19*_jurtina_* and 21*_jurtina_* were not detected. A possible explanation is that the inversions in chromosomes 4*_jurtina_*, 7*_jurtina_*, 8*_jurtina_*, 21*_jurtina_*, 23*_jurtina_*, 25*_jurtina_* and Z*_jurtina_* were fixed in evolution, while the inversions in chromosomes 19*_jurtina_* and 21*_jurtina_* are not fixed and may even have been lost in *E. aethiops*.

## 4. Discussion

### 4.1. Karyotypes of Maniola and Erebia Butterflies

The karyotype of *M. jurtina* was first studied by Federley [59] by using a standard microscopic technique for butterflies from Finland. Federley established that in the first metaphase of meiosis, both males and females have 29 bivalents (*n* = 29). This count of the haploid number (*n* = 29) was then confirmed by the studies by Lorković for Zagreb (Croatia) and Fontainebleau (France) [60] and Bigger for England [61].

The karyotype of *E. ligea* was also previously studied by Federley [59]. For butterflies from Finland, Federley identified 29 chromosome units in oogenesis at the stage of the first metaphase of meiosis (*n* = 29) [59]. In a population from Hokkaido (Japan), known as the subspecies *E. l. rishirizana*, Saitoh and Abe [62] found 28 bivalents in male meiosis and 56 chromosomes in male mitosis, that is, one pair of chromosomes fewer than in Finland. The chromosome-scale genome assemblies confirm the haploid number *n* = 29 for both species [51,52]. They also provide the first information on female heterogamety and sex chromosome systems in these species. *M. jurtina* has 28 pairs of autosomes, as well as sex chromosomes ZZ in males and ZW in females. In *E. ligea*, the whole genome was assembled for the male only. The male chromosome set has 28 autosomes and a pair of ZZ sex chromosomes.

The haploid number *n* = 29 is predominant in the subfamily Satyrinae [28], to which the genera *Maniola* and *Erebia* belong. As a modal number, *n* = 29 occurs in the vast majority of genera of the Palearctic Satyrinae [28,63,64]. The Neotropical satyrines, particularly the basal species, also have *n* = 29 as a weak modal number [65]. The African satyrine taxa have a strong modal *n* = 28; however, *n* = 29 also occurs in a few genera [65,66,67,68,69]. Based on the available data, it can be assumed that *n* = 29 (and, less likely, *n* = 28) is the ancestral number of chromosomes for the subfamily Satyrinae as a whole. The age of the subfamily is about 65 million years [70], during which this karyotype was preserved. Thus, in general, we can speak about the stability of chromosome numbers as one of the main trends in the karyotypic evolution of the Satyrinae.

Against the background of this stability, the species-rich satyrine genus *Erebia* is very unusual. Within this genus, haploid chromosome numbers vary from *n* = 7 in *Erebia aethiopellus* [49] to *n* = 51–52 in *E. iranica* [71]. Accordingly, the expected range of variability in diploid numbers is from 2*n* = 14 to 2*n* = 104, although care must be taken when interpreting the number of chromosome elements in meiosis in terms of diploid numbers. This is due to the fact that in normal meiosis in butterflies, multivalents and univalents can occur in meiosis, for example, as a consequence of the Z0 sex chromosome determination mechanism [32] or the presence of multiple sex chromosomes [34,35], as well as a consequence of polymorphism for chromosomal fusions and fissions [41,72].

The karyotype of *E. aethiops* was first studied by Lorković [60], who, based on the study of a few metaphases in a single male from Croatia, established the haploid number of *n* = 21. The chromosome-scale genome assembly shows that, in Scotland, this species has 18 pairs of autosomes and a pair of sex chromosomes: ZZ in males and ZW in females [53]. Our analysis shows that the Z chromosome of *E. aethiops* resulted from the fusion of the ancestral Z with an autosome and is thus the neo-Z chromosome [2,73]. This neo-Z chromosome is the largest in the set, and the tiny W chromosome is the smallest element in the set. Genomic data (Appendix A) show that the W chromosome of *E. aethiops* is 12 times smaller than the Z chromosome and 5 times smaller than the smallest (18th) autosome.

### 4.2. Dynamic Evolution of Holocentric Chromosomes in Satyrine Butterflies

Due to the extremely high level of interspecific karyotypic differentiation, the genus *Erebia* has become a model for studying rapid karyotypic evolution and the role of chromosomal changes in speciation and post-speciation evolution [74,75]. In this regard, it can be compared with another group of butterflies, the sister genera *Polyommatus* and *Lysandra* (Lycaenidae), in which there is an even higher level of interspecific differences in chromosome numbers [18,24,25,26,76,77]. For both *Erebia* and *Polyommatus*, attempts were made to identify the patterns, trends and possible mechanisms of karyotypic evolution using comparative phylogenetic analysis. In both cases, chromosome numbers were used as traits (proxy features for karyotype) for the analyses [74,75,76,77]. This approach has a serious limitation: although the decrease and increase in the number of chromosomes depend on chromosome fusions and fissions, the same chromosome numbers do not mean that the karyotypes are identical. The chromosome number itself does not provide information on how the whole karyotype originated or how chromosome complements of different species are related [6]. Robust evolutionary analysis should be based on a library of chromosomal rearrangements that differentiate the species under comparison. In our research, we took the first step toward creating such a library.

The comparison between *M. jurtina* and *E. ligea* revealed a high level of macrosynteny in all 29 chromosomes. We did not find any interchromosomal rearrangements (for example, translocations) between these species. Moreover, all of these chromosomes are mostly collinear: that is, they retain the same order of the studied chromosomal blocks. The exceptions are nine small inversions (eight in autosomes and one in the Z chromosome) that accumulated between these taxa over 30 million years of their independent evolution [78]. Thus, in the karyotype evolution of the *Maniola*–*Erebia* lineage, an element of conservatism is observed, that is, the absence of large interchromosomal rearrangements, resulting in the preservation of the number of chromosomes. This conservatism is combined with a more dynamic repatterning of the gene order due to inversions.

A high level of macrosynteny does not necessarily indicate a complete absence of structural chromosomal evolution. It is known that homologous chromosomes can be differentiated through the accumulation of microarrangements, such as insertions and deletions [45,79]. Our data show that, against the background of the conservation of macrosynteny and collinearity, the compared species differ greatly (from 402 to 506 Mb, Table 1) in the total genome size. Most likely, these differences arose as a result of microinsertions and microdeletions of DNA repeats, leading to karyotype divergence, but not changing the gene order.

The evolution of karyotypes was more rapid in the lineage *E. ligea*–*E. aethiops*. The two species split about 12 million years ago [80]. A relatively rapid decrease in the number of chromosomes in *E. aethiops* occurred due to simple fusions, and we did not find any other large intrachromosomal rearrangements in autosomes. The evolution of the sex chromosome was more complex and included, in addition to fusion, the Z chromosome inversion. This finding is highly compatible with previous observations that, in Lepidoptera, sex chromosomes show a more dynamic structural evolution than autosomes [81,82,83,84,85,86,87]. Sex chromosomes are known to play a special role in the formation of reproductive barriers between species [88,89]. They evolve faster than autosomes [90,91], resulting in hybrid sterility genes that are preferentially localized on sex chromosomes [88,89]. We hypothesize that this exceptional role of the Z chromosome may be enhanced through sex chromosome–autosome fusions and Z chromosome inversions, i.e., via rearrangements that are known to promote reproductive isolation between nascent and closely related species [92].

### 4.3. Inversions and Holocentric Model of Chromosomal Speciation

The standard model of holocentric chromosomal speciation [19,93] is based on two main ideas. The first idea is that the holocentric nature of chromosomes contributes to the emergence and subsequent fixation of chromosomal fusions and divisions [3,7,20,75]. The second idea is that chromosomal fusions and fissions facilitate the divergence of incipient species through hybrid sterility and/or recombination repression mechanisms [19,94].

Here, based on our findings, we argue that not only fusions and fissions but also inversions play a significant role in holocentric-chromosome-based speciation. As our data show, inversions are the most common macroscale rearrangements in satyrine butterflies. The role of inversions in the origin and maintenance of reproductive isolation through a recombination suppression mechanism is well documented [92,95]. However, in monocentric organisms, the fixation of widespread paracentric inversions can be difficult. In a heterozygote for paracentric inversion, crossing over within the inverted region leads to the formation of an acentric chromatid and a dicentric chromatid. Both recombinants face problems. The acentric chromatid may be lost, and the dicentric recombinant generates a dicentric bridge during anaphase [2,92]. This problem is absent in holocentric organisms, providing an additional opportunity for karyotype evolution and the formation of new species.

## 5. Conclusions

We conclude that chromosomal evolution is dynamic in satyrine butterflies, even in the lineage that preserves the ancestral haploid chromosome number, *n* = 29. We hypothesize that the exceptional role of Z chromosomes in speciation may be further enhanced by inversions and sex chromosome–autosome fusions. Based on our analysis, we argue that not only fusions and fissions but also inversions are drivers of the holocentromere-mediated mode of chromosomal speciation.

## Figures and Tables

**Figure 1 genes-14-00437-f001:**
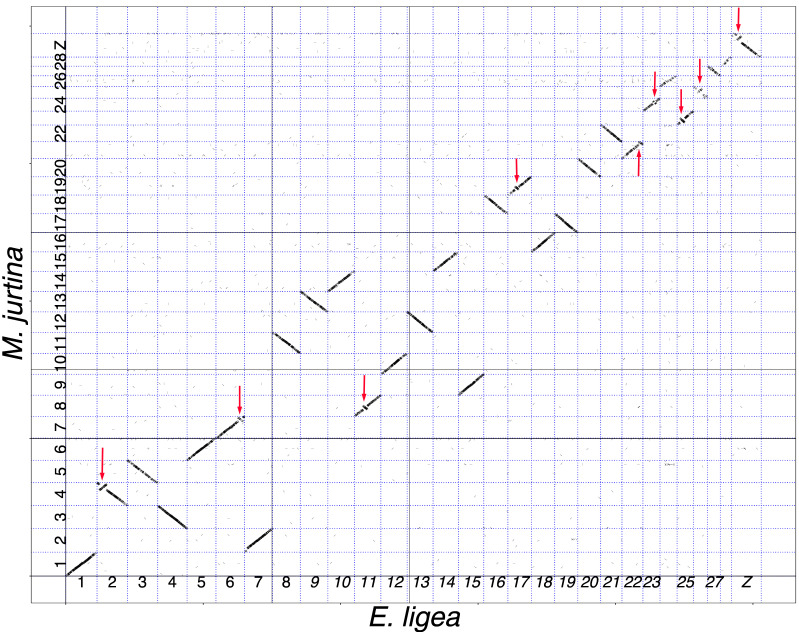
Genome of *E. ligea* (horizontal axis) plotted against genome of *M. jurtina* (vertical axis). It can be seen that chromosome 1*_ligea_* corresponds to chromosome 1*_jurtina_*, chromosome 2*_ligea_* corresponds to chromosome 4*_jurtina_*, chromosome 3*_ligea_* corresponds to chromosome 5*_jurtina_*, chromosome 4*_ligea_* corresponds to chromosome 3*_jurtina_*, and so on, and chromosome Z*_ligea_* corresponds to chromosome Z*_jurtina_*. Each short stroke represents a single syntenic block (a supported alignment), positioned according to its position in the genomes of the compared species. All compared chromosomes are syntenic. Twenty pairs of chromosomes are collinear. Nine pairs of chromosomes show the presence of inversions (shown by red arrows).

**Figure 2 genes-14-00437-f002:**
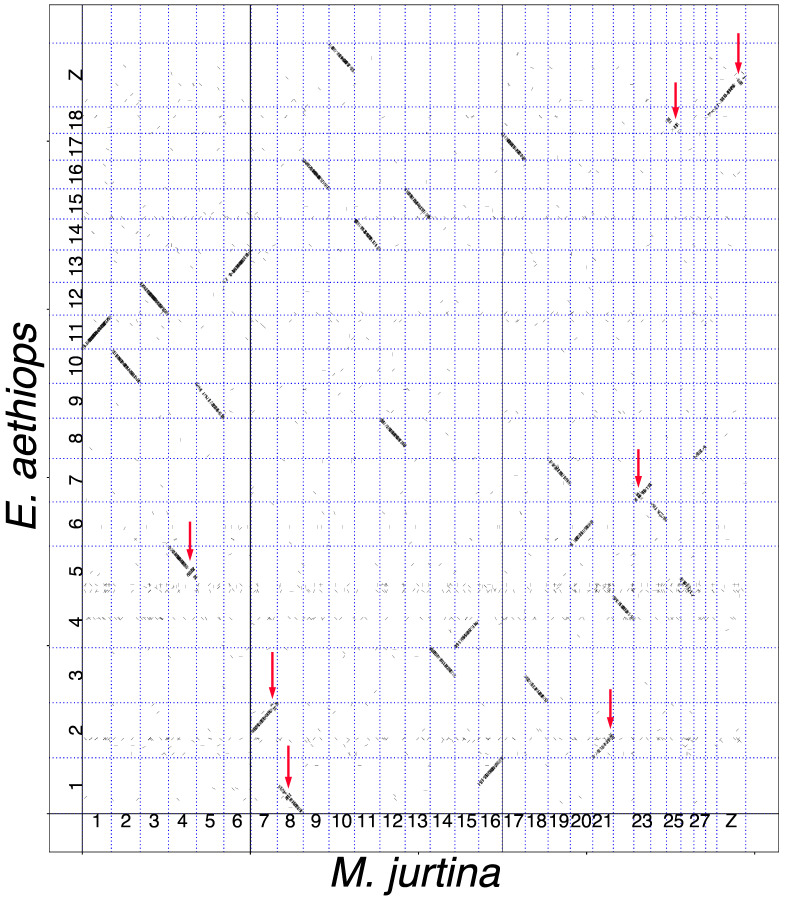
Genome of *M. jurtina* (horizontal axis) plotted against genome of *E. aethiops* (vertical axis). It can be seen that chromosome 1*_jurtina_* corresponds to chromosome 11*_aethiops_*, chromosome 2*_jurtina_* corresponds to chromosome 10*_aethiops_*, chromosome 3*_jurtina_* corresponds to chromosome 12*_aethiops_*, chromosome 4*_jurtina_* corresponds to chromosome 5*_aethiops_*, and so on, and chromosomes Z + 10*_ligea_* correspond to chromosome neo-Z*_aethiops_*. Each short stroke represents a single syntenic block (a supported alignment), positioned according to its position in the genomes of the compared species. All compared chromosomes are syntenic. Seven pairs of chromosomes show the presence of inversions (shown by red arrows).

**Figure 3 genes-14-00437-f003:**
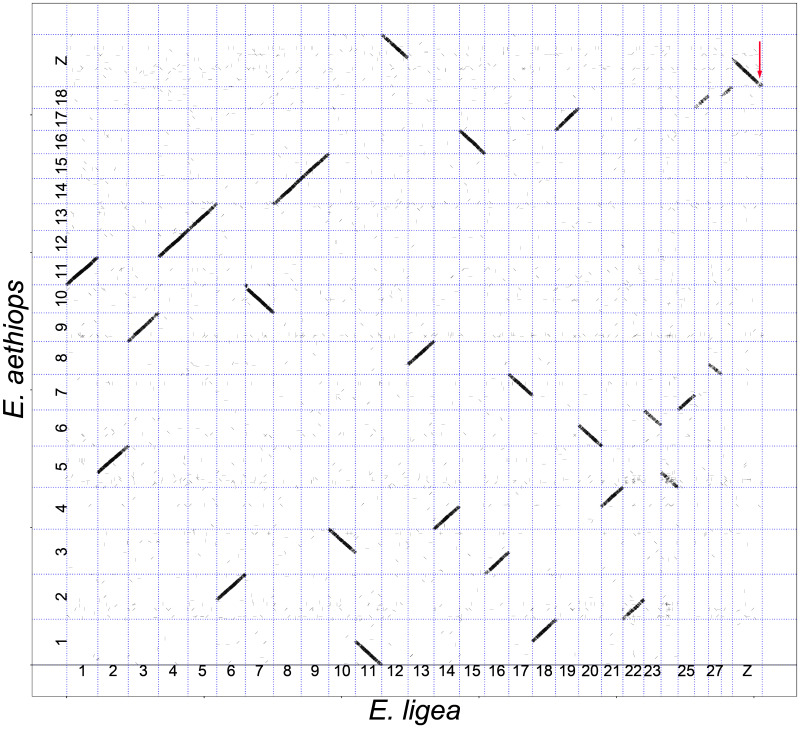
Genome of *E. ligea* (horizontal axis) plotted against genome of *E. aethiops* (vertical axis). It can be seen that chromosome 1*_ligea_* corresponds to chromosome 11*_aetiops_*, chromosomes 2 + 24*_ligea_* correspond to chromosome 5*_aethiops_*, chromosome 3*_ligea_* corresponds to chromosome 9*_aethiops_*, chromosome 4*_ligea_* corresponds to chromosome 12*_aethiops_*, and so on, and chromosomes 12 + Z*_ligea_* correspond to chromosome neo-Z*_aethiops_*. Each short stroke represents a single syntenic block (a supported alignment), positioned according to its position in the genomes of the compared species. The red arrow indicates a small terminal inversion in the Z chromosome.

**Figure 4 genes-14-00437-f004:**
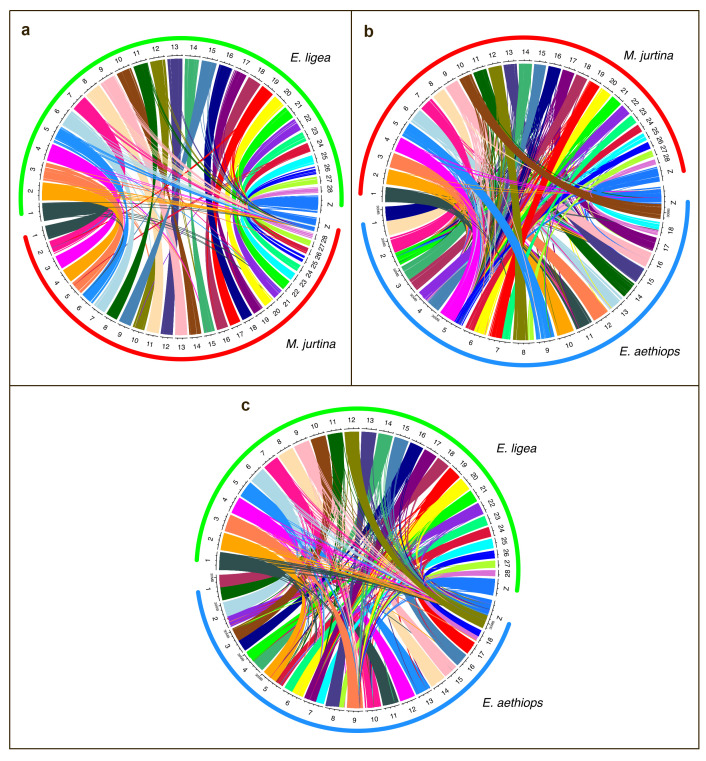
Circos plot showing synteny comparison between *E. ligea* and *E. aehtiops* (**a**), *E. ligea* and *M. jurtina* (**b**) and *E. aethiops* and *M. jurtina* (**c**). Each link corresponds to a single syntenic block (alignment) and is colored by *E. ligea* (**a**,**b**) and *M. jurtina* (**c**) chromosomes. Chromosomes 1–8, 18 and neo-Z of *E. aethiops* were formed as a result of interchromosomal fusions. Chromosomes of *E. ligea* and *M. jurtina* are syntenic.

**Figure 5 genes-14-00437-f005:**
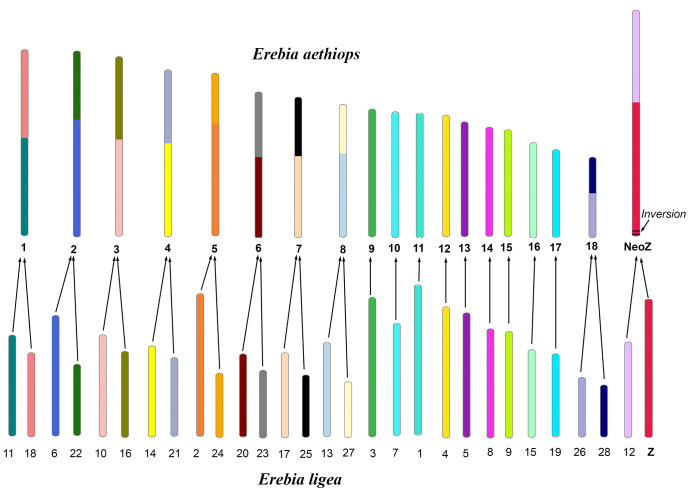
Chromosome comparison between *E. ligea* and *E. aethiops.* Karyotypes of *E. ligea* (*n* = 29) (lower row) and *E. aethiops* (*n* = 19) (upper row) are differentiated by 10 chromosomal fusions and a single terminal inversion in the Z chromosome. The serial numbers of chromosomes of each species are taken from GenBank (Appendix A) and correspond to their sizes.

**Table 1 genes-14-00437-t001:** GenBank IDs, karyotypes, sex chromosome systems and genome sizes in the studied species of the subfamily Satyrinae (according to GenBank data and published data [51,52,53]).

Species	Assembly ID	Diploid Chromosome Number and Sex Chromosomes in Females	Diploid Chromosome Number and Sex Chromosomes in Males	Total Genome Size
*M. jurtina*	ilManJurt1.1	2*n* = 56 + ZW	2*n* = 56 + ZZ	402.0 Mb
*E. ligea*	ilEreLige1.2	unstudied	2*n* = 56 + ZZ	506.4 Mb
*E. aethiops*	ilEreAeth2.2	2*n* = 36 + ZW	2*n* = 36 + ZZ	473.4 Mb

**Table 2 genes-14-00437-t002:** Properties of pairwise alignments (i.e., syntenic blocks) of the Satyrinae whole-genome assemblies after filtration.

Comparison	Maximum Block Length, bp	Sum of Block Lengths, bp	Number of Blocks
*M. jurtina–E. ligea*	13,739	32,722,763	15,870
*E. ligea–E. aethiops*	52,045	67,667,013	30,566
*M. jurtina–E. aethiops*	15,305	16,426,952	16,357

## Data Availability

All of the analyzed chromosome-level assemblies are available via the GenBank links provided (Appendix A).

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
