# Peer review of "Whole-Genome Analysis Reveals the Dynamic Evolution of Holocentric Chromosomes in Satyrine Butterflies"

_genes, 2023, doi:10.3390/genes14020437_

Round 1

Reviewer 1 Report

Congratulations for the MS, it is so interesting, however I have some things I would like to "discuss".

Minor comments:

Line 94. As you mention first Manduca, could it be possible to switch and write first Sphingidae?

Line 335-336. This is the standard example for acknowledgments, please provide yours if it is the case.

Major comments:
In the result section 3.1 it is said: "Karyotypes, genome sizes and DNA repeats". I have some doubts about your contribution here. Did you karyotype any of the studied species? Do you have any information about DNA repeats (there is nothing said there)? How have the genome sizes been measured? by flow cyto? are those sizes from the assemblies?

Why you did not compare M. jurtina vs A. aethiops?

In the figure 3, I would recommend to use the circos plot tool to plot what you want, this is a super good way to visualize the chromosome fusions and inversions. 

Author Response

We greatly appreciate the comments and suggestions of the reviewer as well as the opportunity to make changes and corrections and to submit the revised manuscript.

Our point-by-point responses to the critical comments are here.

Line 94. As you mention first Manduca, could it be possible to switch and write first Sphingidae?

Response: corrected.

Line 335-336. This is the standard example for acknowledgments, please provide yours if it is the case.

Response: corrected.

In the result section 3.1 it is said: "Karyotypes, genome sizes and DNA repeats". I have some doubts about your contribution here. Did you karyotype any of the studied species? Do you have any information about DNA repeats (there is nothing said there)? How have the genome sizes been measured? by flow cyto? are those sizes from the assemblies?

Response: We thank the referee for this important comment. This information was obtained from GenBank. In fact, it represents the material that was used for our analyses. Therefore, we moved this subsection (including Table 1) to the Materials and Methods section.

Why you did not compare M. jurtina vs A. aethiops?

Response: In the revised MS we provide the comparison between M. jurtina and E. aethiops genomes.

In the figure 3, I would recommend to use the circos plot tool to plot what you want, this is a super good way to visualize the chromosome fusions and inversions.

Response: In the revised MS we added the circus plot analysis of the genomic data to visualize the synteny and chromosomal rearrangements.

Reviewer 2 Report

The present paper presented by Pazhenkova and Lukhtanov, deals with a very interesting topic about holocentric chromosomes and speciation.

The paper is well written, introducing the subject, their hypothesis and how results leads to the final discussion. However, I have some minor comments to the authors.

Please, add the reference to the chromosome numbers of the species analyzed in this work. On Table 1 and results (lines 139-140), as you mention on discussion.

As recommendation to authors, I would make some changes in dotplot figures (1 and 2). I would erase the black lines on the grid which do not correspond to chromosome limits. For example, on fig 1, the 5th chromosome is divided by a black line. This makes the figure difficult to read. Other change is to better locate the X and Y axis reference, outside of the grid. Or limit the grid up to the lines of the axes. But this point is merely aesthetic.

Figure 1, please modify the caption (line 186). Supplant dots by "and so on" for example. Same for figure 2 on line 205, the dots must be changed by words.

Author Response

We greatly appreciate the comments and suggestions of the reviewer as well as the opportunity to make changes and corrections and to submit the revised manuscript.

Our point-by-point responses to the critical comments are here.

Please, add the reference to the chromosome numbers of the species analyzed in this work. On Table 1 and results (lines 139-140), as you mention on discussion.

Response: This information was obtained from GenBank. In fact, it represents the material that was used for our analyses. Therefore, we moved this subsection (including Table 1) to the Materials and Methods section.

As recommendation to authors, I would make some changes in dotplot figures (1 and 2). I would erase the black lines on the grid which do not correspond to chromosome limits. For example, on fig 1, the 5th chromosome is divided by a black line. This makes the figure difficult to read. Other change is to better locate the X and Y axis reference, outside of the grid. Or limit the grid up to the lines of the axes. But this point is merely aesthetic.

Response: we corrected the dotplot figures as recommended by the reviewer.

Figure 1, please modify the caption (line 186). Supplant dots by "and so on" for example. Same for figure 2 on line 205, the dots must be changed by words.

Response: we modified the caption as recommended by the reviewer.

Round 2

Reviewer 1 Report

The authors have address al the comments. 

Thank you! I really appreciate this MS